# Is Family Structure Associated with Deviance Propensity during Adolescence? The Role of Family Climate and Anger Dysregulation

**DOI:** 10.3390/ijerph17249257

**Published:** 2020-12-10

**Authors:** Valeria Saladino, Oriana Mosca, Marco Lauriola, Lilli Hoelzlhammer, Cristina Cabras, Valeria Verrastro

**Affiliations:** 1Department of Human, Social and Health Sciences, University of Cassino and Southern Lazio, 03043 Cassino, Italy; 2Department of Education, Psychology, Philosophy, University of Cagliari, 09123 Cagliari, Italy; oriana.mosca@unica.it (O.M.); ccabras@unica.it (C.C.); 3Department of Developmental and Social Psychology, University of Rome “Sapienza”, 00185 Rome, Italy; marco.lauriola@uniroma1.it; 4Department of Philology and Literature, LMU Munich, Ludwig-Maximilians-Universität, 80539 Bavaria, Germany; Lilli.Hoelzlhammer@campus.lmu.de; 5Department of Medical and Surgical Sciences, University of “Magna Graecia”, 88100 Catanzaro, Italy; valeriaverrastro@unicz.it

**Keywords:** family system, adolescence, deviance, aggressive behavior, anger dysregulation

## Abstract

Transgressive conduct and opposition towards the rules often characterize adolescence. During the development, antisocial and aggressive behavior could be a way to grow personally and to be independent. According to previous studies results, the family has a high impact on teens’ aggressive behaviors and moral disengagement. Our research involved 2328 Italian adolescents (13–19 years old) who have filled in the following questionnaires: deviant behavior questionnaire; aggression questionnaire; family communication scale; moral disengagement scale; the multidimensional scale of perceived social support. Our study investigated the role of family structure on deviance propensity through family climate and anger dysregulation joint influence. We conducted a mediation analysis to reach this goal using structural equation modeling (SEM). We have also conducted a multigroup analysis in order to evaluate gender differences in the SEM. Results showed that both family climate and anger dysregulation mediated the relationship between family structure and deviance propensity. The multigroup analysis revealed that the indirect relationship between variables through family climate is significant for both boys and girls (higher in females); variables indirect relationship through anger dysregulation was significant only for girls. These data could be useful for prevention and intervention programs on children–parent relationships and to reduce antisociality and teenager’s aggressive behavior.

## 1. Introduction

### 1.1. Normative or Deviant? Personal and Social Identity during Adolescence

Deviance concept is strongly associated with a specific historical period social and cultural norms; these norms belong to all people who play a role in the social context. From this point of view, the term “to divert” means breaking the social agreement and the established rules and norms [1]. De Leo and Patrizi [2] describe deviance as a social construct influenced by personal, environmental, and family factors and also by the decision-making process based on individuals’ social and behavioral elements interaction [3]. Deviating the norms involves a transgression or adherence to a certain social role, which may be positive or negative. This role can influence prejudice and social stigma [4,5].

According to social psychology, the identity concept characterizes individual growth [6]. The process of identity formation reaches its climax during adolescence with the beginning of the most demanding social experiences. These are characterized by the reconsideration of interpersonal relationships with parents, and a consequent questioning of rules and behaviors learned during childhood [7]. Additionally, adolescents performing criminal conduct may experience negative expectations from others [8,9]. These expectations alter and shape the identity and image of an individual in the way of a “self-fulfilling prophecy”: by accepting the negative expectations that others have about them, they start to become exactly like the negative image that is projected onto them [10]. On the contrary, positive expectations favor identity growth in teens [11]. Therefore, society, groups, and family continually confirm a person’s sense of identity through positive or negative reinforcements such as gratifications, frustrations, judgments, and evaluations [12]. As suggested by the labeling theory [13], the individuals recognize themselves as persons with negative social values attributed by others, assuming a conforming identity. In the case of antisocial and deviant conduct, these individuals could develop and maintain deviant or aggressive identity over time [14,15]. Labeling teenagers as delinquents or criminals increases the risk of an antisocial identity and a criminal career, establishing a deviant role [16,17].

During adolescence, it is important to make a difference between deviant behavior as delinquency form and deviant behavior as rebellion or distress form [18]. This period is characterized by sudden changes (need for autonomy or need for closeness), conflicts and insecurities that can involve transgressive behaviors and rebellion [19]. According to De Leo et al. [2], there are two forms of deviances: (1) contingent deviance (belonging to growth) involves transgressive actions and behaviors; it ends during adulthood; it is a normative deviation; (2) persistent deviance is associated with a behavioral modality which leads individuals towards criminality, affirming over time in a criminal career.

A deviant and aggressive behavior could be a distress expression, through which teenagers communicate their negative emotions and feelings such as confusion and interpersonal difficulties [20]. For adolescents with a strong sense of unease, not the action, but the underlying motivation contains meaning. Hence, it is difficult to define teenagers’ illegal and antisocial acts as conscious choices because they do not think about the consequences [21]. Teenagers’ aggressive attitudes can lead to behavioral, cognitive and emotional anger dysregulation, and it may be externally or internally expressed [22]. In the first case (common to males), teenagers tend to physically attack; they use violence [23]. In the second case (common to females), aggression is internalized, and it is expressed through anger and hostility [24]. The aggressive and antisocial attitude in adolescence is made possible by cognitive strategies called mechanisms of moral disengagement. They can be considered directly proportional to the aggressive act and bridge the gap between action and thought when people act morally unjust or are exposed to judgment [25]. In adolescence, the mechanisms of moral disengagement have a dual function: to justify actions considered wrong by others and to affirm one’s autonomy and independence [26].

### 1.2. Adolescence, Family Structures, and Climate

According to the psycho-sociological perspective, adolescents’ distress derives from a perceived problem in the relationship between an individual and the context [27]. If neither family nor institution is interested in adolescents’ distress, it may become chronic and involve behavioral and psychological problems, such as addiction, delinquency, mental and physical diseases [28]. As mentioned before, deviance and aggressive behavior can assume different meanings during adolescence than in adulthood. For this reason, it is important to analyze them according to a developmental perspective. In this regard, the family system’s importance has been emphasized by several studies focusing on the large impact of family communication and family structure in adolescents’ behavior and well-being on different levels [29,30,31]. Indeed, there are some theoretical perspectives, which could explain family structure differences in adaptation. For instance, society stigmatizes adoptive families the most due to their lack of biological ties and their authenticity of the parent’s role [32]. For this reason, teenagers belonging to these families often suffer from discrimination in social and institutional contexts. Correspondingly, these teenagers should have a lower level of well-being and worse relationships with family members than those who live in other types of families. There should be no difference between adoptive and original families since both parents equally protect their children from developing academic and social adjustment problems [33,34]. Single-parent, step-parent and divorced families may lead to delinquent behavior, substance abuse and externalizing problems [35,36,37,38,39]. However, researchers have shown conflicting results about adoptive families’ influence. On one hand, adoption seems to be connected to academic and behavioral issues [40,41,42]; on the other hand, other studies found no significant differences, showing a similar level of warmth, communication, family climate, and support in adopted and nonadopted children [43]. Furthermore, comparing the quality of family relationships and well-being in two-parent biological families, single-mother families, and stepfather and stepmother families, results showed that two-parent biological families report fewer children externalizing problems than adoptive ones, but a higher level of conflict with children than stepmother and stepfather families. In addition, mothers invest the same time in children regardless of the family type, whereas fathers in two-parent biological families spend more time with their children than in other family structures. These results underline the importance of evaluating family structure. It seems important to take into account family climate and processes that characterize all the family structures differently since they could affect behavior and especially deviant and risky conducts in several ways [44]. Most of the time, adolescents’ perception of closeness to parents should affect their behavior positively. For instance, the analysis of adolescents’ focus group data regarding their perceptions of violence and coping strategies with conflict management demonstrated that participants who perceived their parents as more conscientious and present were less likely to engage in delinquent acts [45].

The same considerations can be applied to the impact of support and good family communication and its effects on adolescent behavior [46,47]. Several studies showed that parent–child disclosure and mutual trust lead to support and good family communication [48,49]. This openness facilitates emotional management, improves the ability to react to stressful events, and reduces delinquent acts, promoting greater well-being [44,46]; on the contrary, poor family communication and low support involve an affective and emotional gap that can lead to physical violence and aggressive behavior [45,46,47]. Overall, the negative quality of family climate is strongly related to deviance and antisocial conduct [49]. Moreover, family communication is related to self-control and self-efficacy sense which promotes self-esteem and positive behavior [50,51]. Studies also underline parents and children relationship according to gender [52,53]. Specifically, the data showed a greater tendency of females to open up in a family context, especially with the mother [54]. On the other hand, the father-daughter relationship has not been explored as much. However, it seems that both males and females perceive fathers as strong and perfect figures, mentors to be imitated to build their own identity [51]. Overall, the mother relationship represents a socialization factor since it provides mental and emotional support [53]; instead, the father teaches values, rules and moral obligations through concrete attitudes. Parents’ trust and empathy bond represent a protective factor for future relationships, and it affects teenagers’ behavior [54]. However, the involvement of both parents in their children’s lives is the most favorable situation. Parents’ presence and support are associated with a good family climate and with teenager’s growth adjustment [49].

### 1.3. Family Structure and Deviance in Adolescence: An Italian Perspective

The Italian family is changing; the Italian National Institute of Statistics (Istat) has shown a decline in marriages and an increase in divorces [55]. People do not get married; the marriage is shorter, and there are many extended families and single-parent families [56]; this leads to teenagers’ behavioral and emotional change [57]. Italian teenagers living in reconstituted families usually behave negatively, and they experience emotional distress [58]. However, the influence of a family’s socioeconomic status, its functioning, and the parent’s behaviors and psycho-physical well-being are not well researched in Italy. It should be studied in-depth, distinguishing the effects of the family structure according to the specific behaviors and well-being aspects of the adolescent [59].

The representations in which children perceive themselves as members of a family and experience family dynamics have changed. Young people’s space of freedom has become greater than in the past; they have more opportunities in terms of physical and mental movement. This, however, leads also to an “empty space” due to the crisis in the parental educational role. Mostly, the risk derives from difficulties in family members’ relationships. The strong confusion of roles between parent and children lead to family instability. This state of uncertainty may also instigate risky and deviant behavior. The juvenile court reported the following conduct problems spread among adolescents from disrupted families: early school drop-out, early onset in sexuality [60,61], bullying, vandalism, wandering, alcohol use, gambling, belonging to a criminal environment, drug abuse and dealing, sexting, sexual abuse, violent and aggressive acts against parents. The analysis of justice-involved Italian adolescent’s family context pointed out a lack of educational coherence, as the parent’s attitudes varied between excessive severity and permissiveness [62]. Especially families characterized by open and short-term cohabitations between a biological parent and a stepmother or stepfather, with whom the biological parent is having a sexual relationship and does not share a stable affection, turn out to be criminogenic [63].

Despite these pieces of evidence, little is known about the relationship between family structure and deviance propensity in Italian adolescence. Future research is needed to analyze the phenomenon in-depth and for comparison with main international results.

### 1.4. Socioeconomic Status, Family Structure, and Risky Behavior in Adolescence

Another important aspect, which influences adolescents’ well-being and behavior is socioeconomic status (SES) [64]. The SES is a socio-environmental factor that indicates the condition in which an individual or a family lives. Several studies have shown that a high SES has a positive effect on the outcomes of school achievements during childhood and adolescence [65,66,67,68]. In contrast, a low SES has a negative influence on teenagers’ behaviors (aggression, use of drugs, delinquent behaviors) [69,70]. According to the Yonkers project evaluation, adolescents who live in a low-SES context are more likely to be involved in risky conduct, such as becoming substance users [71]. In addition, a report on criminal activity based on criminal-offender records from the Maryland Department of Justice found that adolescents who moved away from low poverty conditions are less likely to be arrested for violent crimes than their peers who remain in public housing or low-SES context [72]. An economically stressed family could also expose adolescents early on to criminality or deviance propensity. Damm and Dustmann [73] investigated this effect on a sample of 0 to 21-year-olds allocated randomly to disadvantaged neighborhoods with their families. Results showed a stronger tendency to criminality and the development of deviant behavior in these adolescents, also due to the low level of parental control and support [74,75]. The age of exposure has been found to be the most important aspect. In Damm and Dustmann’s study, the participants were allocated to economically disadvantaged neighborhoods at different ages: 0–5, 6–9, 10–14, and 15–21 years of age. Results showed that those between 10 and 14 years old during the assignment are most affected by the economically disadvantaged neighborhoods in comparison to the other groups. Moreover, the economic disadvantage also affects the probability of obtaining a higher education and or employment. It is interesting to note the gender difference in these results. Prior research showed that gender differences are not due to the influence of different social factors but that males and females are affected differently by the same condition [76,77]. In a study of 400 economically disadvantaged cities, Heimer et al. [76] found that in some cases, females were more likely to be involved in arrests, especially property crime, than males. Thus, economic disadvantage moderates the effect of gender on offending. According to the disorganization theory by Shaw and McKay [77] disadvantageous and economically distressed neighborhoods with ineffective social control by family and community provided more opportunities for adolescents to be affiliated with violent and deviant peers. This lack of control has a stronger impact on females rather than on males because females are more guided by the family and more involved in conventional activities in their home, school, and community. When social control breaks down, females are more likely exposed to risky behaviors and crimes, as reported by data from the Project of Human Development in Chicago Neighborhoods (PHDCN) [78].

SES also has a strong connection with family structure. Research evidence showed the link between low SES and single-parent families, especially single-mothers, who seem to be more affected by economic problems [79]. Indeed, original families had more economic resources and lower poverty rates than single-mother families, in particular never-married mothers. In addition, mother-partner families reported higher incomes and lower poverty rates than single-mother families [80]. Mostly in two-parent families (original and step-parent), there is a balance among partners in working hours and children’s education and support, while single mothers work longer than those in original two-parent families. This negatively affects the support and communication between mothers and children [81]. The situation is different for single fathers, who have lower income than those who are married but are still better off than single mothers. Furthermore, single fathers reported having a higher rate of cohabitation than single mothers, and this cohabitation guarantees better education and support, even if it is not the original family [82].

Good economic resources affect parent–children relationships, not only in the promotion of better school education, health care, and opportunities but especially in the quality of the relationship with parents. Economically stressed families cannot provide emotional and parental resources like those who live in optimal economic conditions [83]. Therefore, the economic resources are indirectly connected to the family structure, as regarding the investment of time and money. This connection could explain poorer outcomes for children who live in single-parent families and stepfamilies. Several studies [80,81] investigated the disadvantages associated with divorce and single parenthood, finding that children receive less parental time and support in both single-parent families and stepfamilies. On one hand, single-parent families exercise less control over their children than two parents-families; on the other hand, stepparent-families provide lower support and disclosure than original-families [84]. Findings on parental control in never-married and ever-married mothers suggested that the control is affected by the number of adults who take care of children rather than by single-mothers’ skills. These results propose that parental behaviors can compensate or increase the effects of any economic difference between types of family structure and that there is a reciprocal influence between economic resources, parenting, and family structure. Overall, low income linked to poor control and support is associated with single-parent families, especially single-mothers [85]. Two-parent families (original or adoptive) guarantee more support and communication in general, but only original families seem to be a protective factor in not developing behavioral problems [80].

Taken together, these findings suggest that parental relationship, communication, support, family structure, and economic resources can influence teen’s antisocial and aggressive behaviors, emphasizing the importance of parental skills to decrease developmental risks for adolescents. Many of these results are well-known, but there are not many studies on aggression and antisocial conduct on Italian adolescents. The aims of this study are: (a) identifying the role of family structure on deviance propensity among Italian adolescents through the joint influence of family climate and anger dysregulation; (b) evaluate gender differences in our mediation model; (c) using the main results for future studies to collect data that considers parent–children relationships to gain a better understanding of the causes and motivations behind adolescents’ antisocial behavior. Specifically, we expected that: (H1) an intact (vs. single parent) family structure elicits a positive (vs. negative) family climate; (H2) an intact (vs. single parent) family structure promotes (vs. prevents) anger dysregulation; (H3) family climate and anger dysregulation uniquely predict deviance propensity; specifically, (H4a) family climate protects from the development of deviance propensity; (H4b) anger dysregulation triggers the deviance propensity; (H4c) both family climate and anger dysregulation influence deviance propensity; (H5) gender has a moderating role on all the previous hypothesis (from H1 to H4c); (H6) SES influences all the previous hypotheses (from H1 to H5).

## 2. Materials and Methods

### 2.1. Participants

The participants are 2328 teenagers (58.1% females) (mean age: 16.36 years old) (DS = 1.50; range 13–19) from these Italian regions high schools: Lazio (*n* = 4), Campania (*n* = 4), Sicily (*n* = 3), Emilia-Romagna (*n* = 1), Piemonte (*n* = 1) and Lombardia (*n* = 1). These schools have voluntarily joined the research after an announcement by the University of Cassino and Southern Lazio that asked for the dissemination of the research throughout the Italian territory. The sample is not representative with regard to Italian regions due to the unequal geographic distribution of the participants. Indeed, 25.9% of the sample come from the Center of Italy, 11.7% from the North, and 62.5% from the South. Teens majority live with both mother and father (84.9%); the others only with the mother (13.2%) or the father (1.9%). An interviewer explained the study goals to potential participants and informed them that they could withdraw from the study, and their responses would have remained anonymous. Researchers have obtained informed consent for adolescents, signed by their parents. The study was conducted following the Declaration of Helsinki and approved by the Institutional Review Board of the University of Cassino and Southern Lazio (Italy). Data were collected from November 2017 until December 2018.

### 2.2. Measures

#### 2.2.1. Deviant Behavior Questionnaire

The deviant behavior questionnaire (DBQ) investigates teenagers’ common risks and deviant behaviors. It belongs to the sociodemographic questionnaire, and it has 9 items. It investigates the tendency to commit illegal actions (e.g., “Have you ever stolen anything?”; “Have you ever thought to steal?”; “Have you ever illegally downloaded music or movies?”) and the tendency to exhibit aggressive attitudes (e.g., “Have you ever threatened or assaulted someone with a weapon?”; “Have you ever verbally or physically attacked someone?” “Have you ever been in a fight at school, at the stadium or in a public place?”). The questionnaire was adapted from the international self-reported delinquency study [86]. The scale internal consistency coefficient is 0.69.

#### 2.2.2. Aggression Questionnaire

The aggression questionnaire (AQ) by Buss and Perry [87,88] is a self-report questionnaire that evaluates the tendency to aggression. The questionnaire derives from a factorial analysis carried out on the items of the hostility inventory [89] used to assess the levels of hostility, from which four factors have emerged distributed over 29 items: physical aggression (PA) (e.g., “I’ve been so angry that I destroy things”), verbal aggression (VA) (e.g., “I openly tell my friends when I disagree with them”), anger (A) “When I’m frustrated, I openly show my irritation” and hostility (H) “I know that my “friends” talk about me behind my back”. Assuming that aggressive behavior can be acted upon in a variety of ways and that it develops along a continuum, the authors evaluate the types of aggressive behavior on a Likert scale from 1 to 5 (1 = totally false; 5 = totally true). High scores in each subscale are equivalent to a higher propensity to aggression. Internal consistency coefficients are the following: anger (0.68), physical aggression (0.73), verbal aggression (0.61), hostility (0.74).

#### 2.2.3. Family Communication Scale

The family communication scale (SCF) is a self-report questionnaire published by Ardone and D’Atena [90] from the Italian adaptation of parent–adolescent communication (PAC) [91,92,93]. The questionnaire is composed of 24 items, separately for the mother and for the father, evaluated on a Likert scale from 1 to 5 (1 = strongly disagree; 5 = strongly agree), resulting in a total number of 48 items. Internal consistency coefficients of the scale are the following: Communication with mother (0.81) and communication with father (0.86).

#### 2.2.4. Moral Disengagement Scale

The moral disengagement scale [94] is a self-report that evaluates the homonymous construct identified by Bandura [95,96]. The author identified specific mechanisms of moral disengagement that allow bridging the gap between thought and action, which is created when individuals act against socially recognized moral values. The greater the moral disengagement, the lesser is the sense of guilt and the need to repair the evil caused by the damaging conduct. In its version for adolescents, the instrument is composed of 14 items evaluated on a Likert scale from 1 to 5 (1 = completely false; 5 = completely true). The items of the scale derived from the mechanisms of moral disengagement mentioned: moral justification (e.g., “It is good to use force against those who offend your family”); euphemistic labeling (e.g., “Taking someone’s moped without permission is only a ‘loan’”); advantageous comparison (e.g., “Stealing some money is not at all serious compared to those who steal large amounts of money”); displacement of responsibility (e.g., “If children are not well educated at home, they cannot be blamed if they then behave badly”); distortion of consequences (e.g., “Mocking doesn’t hurt anyone”); dehumanization of the victim (e.g., “Some people deserve to be treated harshly because they don’t have feelings that can be hurt”); attribution of blame to the victim (e.g., “Boys who are treated badly usually deserve it”) and spreading of responsibility (e.g., “Boys can’t be blamed if they use bad words since most of their friends do the same”). The higher the score is reported, the higher is the moral disengagement. The internal consistency coefficient for the scale is 0.87.

#### 2.2.5. Multidimensional Scale of Perceived Social Support

The multidimensional scale of perceived social support (MSPSS) conceived by Prezza and Principato [97] aims to investigate the support perceived by the person on three dimensions: family, friendship, and society. The measure is composed of 12 affirmations investigated on a Likert scale from 1 to 6, which analyze how much one feels understood and welcomed by one’s family, friends, and social network. The thirteenth item is instead open-ended and asks to specify the person defined in the test as “a particular person who is important for you”. For the aims of the present study, we only used the family dimension. The internal consistency coefficient for the family subscale is 0.92.

#### 2.2.6. Socioeconomic Status (SES)

The Hollingshead SES index is an extensively used measure of the socioeconomic status of a child’s family [98]. We obtained the index from a standardized coding procedure of the occupational prestige and educational level of both parents. For professional status, we considered nine levels ranging from 1 (e.g., farm laborers/menial service workers) to 9 (e.g., higher executives and major professionals). For education, we considered seven levels, from 1 (i.e., less than seventh grade) to 7 (i.e., graduate professional training). The Hollingshead SES index is then obtained as a weighted average of professional status and education, using a weight of 5 and 3, respectively. The index ranges from 8 to 66, with higher scores reflecting higher SES. The mean SES score in the present study was 35.86 (SD = 9.48), a value that corresponds to the middle stratum of Hollingshead’s classification of social classes.

### 2.3. Statistical Analysis

Descriptive statistics, zero-order correlations, and preliminary descriptive analyses of variance (ANOVAs) to compare subgroups of adolescents (boys vs. girls) and family structure (intact vs. single parent) were carried out on the measures used in the present study (see Table 1, Table 2 and Table 3). Structural equation modeling (SEM) was used to test the conceptual model outlined in Figure 1. The analyses were carried out using the “lavaan” [99] package for R. The only exogenous variable in the model was an observed dummy for family structure (intact = 0; single parent = 1). Deviance propensity was the primary endogenous latent variable, measured using three composite scores, corresponding to deviant behavior, moral disengagement, and physical aggression (one of the composite scores of the AQ). The latent mediators were family climate (measured using the Perceived Family Support of the MSPSS, and the composite scores of communication with mother and communication with father of the SCF) and anger dysregulation (measured by the composite scores of the AQ: anger, hostility and verbal aggression).

**Figure 1 ijerph-17-09257-f001:**
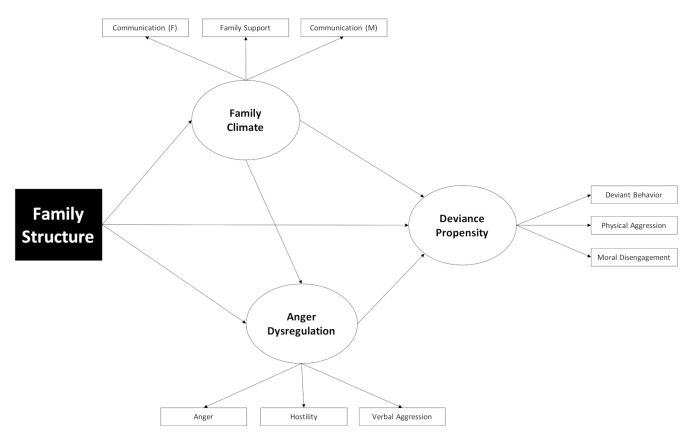
Conceptual model.

The model was fitted using diagonally weighted least squares (DWLS). We first carried out a single group analysis aimed to assess the quality of the measurement model for each latent variable and the feasibility of the proposed indirect relationships (see Figure 2). In particular, the average extracted variance (AVE) is a measure of the amount of variance that was accounted for by a construct relative to the total amount of variance in its indicators. According to Fornell and Larcker [100], an AVE ≥ 0.50 supports the convergent validity of the indicator variables. The model fit was evaluated using the comparative fit index (CFI), Tucker–Lewis index (TLI), root mean square error of approximation (RMSEA), and standardized root mean square residual (SRMR). CFI and TLI values greater than 0.95 support a good fit, with a value above 0.90 deemed acceptable [101]. RMSEA and SRMR, respectively lower than 0.06 and 0.08, support a good fit. The significance of indirect relationships was tested using nonparametric bootstrap confidence intervals with 5000 resamplings. Because the exogenous variable was family structure, the partially standardized indirect effect can be interpreted as the average increase in the deviance propensity scores for adolescents coming from single-parent families, accounted for by the mediators (i.e., family climate and anger dysregulation). Large, medium, and small effect size thresholds are 0.26, 0.13 and 0.02, respectively [102]. The multigroup analysis aimed to test the invariance of the structural paths and indirect relationships. Before testing the invariance of path coefficients between girls and boys, it is necessary to ensure that the latent variables were measured in the same way in both groups. Therefore, we tested the metric invariance of the model. In so doing, we estimated a baseline model in which the factor loadings were free to vary between genders. Next, we constrained the factor loadings to be equal between genders and compared the fit of the constrained to that of the unconstrained one. A non-significant chi-square difference test supports the metric invariance. With large samples, the chi-square difference test may be oversensitive to trivial changes in the factor loadings. According to the standards for establishing measurement invariance of psychometric scales, the following criteria were used (i.e., ΔCFI < 0.010 [103]; ΔNCI < 0.007 [104].

## 3. Results

### 3.1. Descriptive Analysis, Correlations, and ANOVAs

We have calculated means, standard deviations, alphas, skewness, kurtosis, and zero-order correlations between variables (see Table 1).

**Table 1 ijerph-17-09257-t001:** Means, standard deviations, alphas, skewness, kurtosis and zero-order correlations between variables (*n* =2328).

	Minimum	Maximum	Mean	SD	Sk	C	1	2	3	4	5	6	7	8	9
Deviant Behavior	0	9	2.26	1.80	1.07	1.14	1								
Physical Aggression (PA)	1	5	2.42	0.85	0.43	−0.44	0.52 ^**^	1							
Anger (A)	1	5	2.69	1.24	0.04	−0.37	0.34 ^**^	0.54 ^**^	1						
Hostility (H)	1	5	2.98	0.77	−0.13	−0.33	0.18 ^**^	0.32 ^**^	0.59 ^**^	1					
Verbal Aggression (VA)	1	5	3.19	0.71	−0.05	0.17	0.26 ^**^	0.44 ^**^	0.60 ^**^	0.48 ^**^	1				
Moral disengagement	1	5	2.20	0.75	0.62	0.21	0.33 ^**^	0.51 ^**^	0.29 ^**^	0.19 ^**^	0.23 ^**^	1			
Communication with Mother	1	5	3.45	0.55	−1.11	3.06	−0.08 ^**^	−0.01	0.03	0.10 ^**^	0.20 ^**^	0.02	1		
Communication with Father	1	5	3.22	0.65	−0.73	1.47	−0.14 ^**^	0.02	−0.03	−0.02	0.09 ^**^	0.11 ^**^	0.52 ^**^	1	
Perceived Family Support	1	7	5.71	1.38	−1.45	1.70	−0.15 ^**^	−0.11 ^**^	−0.15 ^**^	−0.19 ^**^	0.03	−0.04	0.53 ^**^	0.44 ^**^	1

Notes: ** *p* < 0.01.

We have conducted two preliminary ANOVAs: the first one with gender as a between-subject factor (see Table 2) and the second one with family structure as a between-subject factor (see Table 3).

**Table 2 ijerph-17-09257-t002:** ANOVAS on all the variables examined with gender as between-subject factor.

	Boys (*n* = 976)	Girls (*n* = 1352)				
	M	M	*df*	*p*	F	η_p_^2^
Deviant Behavior	2.86	1.82	(1, 2327)	***	207.71	0.08
Physical Aggression (PA)	2.75	2.19	(1, 2327)	***	278.78	0.11
Anger (A)	2.77	2.87	(1, 2327)	***	10.06	0.04
Hostility (H)	2.86	3.06	(1, 2327)	***	35.79	0.01
Verbal Aggression (VA)	3.15	3.21	(1, 2327)	n.s.	3.41	0.00
Moral disengagement	2.47	2.00	(1, 2327)	***	241.83	0.09
Communication with Mother	3.44	3.46	(1, 2327)	***	1.34	0.00
Communication with Father	3.29	3.17	(1, 2327)	n.s.	20.75	0.01
Perceived Family Support	5.82	5.62	(1, 2327)	***	11.31	0.01

Notes: *** *p* < 0.001; n.s. = not significant.

Boys reported significantly higher levels of deviant behavior (F(1, 2327) = 207.71, *p* < 0.001, partial ηp^2^ = 0.08), physical aggression (F(1, 2327) = 278.78, *p* < 0.001, partial ηp^2^ = 0.11), moral disengagement (F(1, 2327) = 241.83, *p* < 0.001, partial ηp^2^ = 0.09) and perceived family support (F(1, 2327) = 11.31, *p* < 0.001, partial ηp^2^ = 0.01) compared to girls. Girls reported significantly higher levels of anger (F(1, 2327) = 10.06, *p* < 0.001, partial ηp^2^ = 0.04) hostility (F(1, 2327) = 35.79, *p* < 0.001, partial ηp^2^ = 0.01) and communication with mother (F(1, 2327) = 1.34, *p* < 0.001, partial ηp^2^ = 0.01) compared to boys.

**Table 3 ijerph-17-09257-t003:** ANOVAS on all the variables examined with family structure as between-subject factor.

	Intact Structure (*n* = 1868)	Single Parent (*n* = 326)				
	**M**	**M**	***df***	***p***	**F**	**η_p_^2^**
Deviant Behavior	2.19	2.63	(1, 2327)	***	18.19	0.01
Physical Aggression (PA)	2.41	2.56	(1, 2327)	***	9.37	0.00
Anger (A)	2.81	3.00	(1, 2327)	***	22.94	0.01
Hostility (H)	2.96	3.14	(1, 2327)	***	20.27	0.01
Verbal Aggression (VA)	3.18	3.25	(1, 2327)	n.s.	3.06	0.00
Moral disengagement	2.20	2.20	(1, 2327)	n.s.	0.15	0.00
Communication with Mother	3.46	3.40	(1, 2327)	*	4.82	0.00
Communication with Father	3.27	2.91	(1, 2327)	***	62.56	0.03
Perceived Family Support	5.80	5.18	(1, 2327)	***	62.22	0.03

Notes: *** *p* < 0.001, * *p* < 0.05., n.s. = not significant.

Adolescents living with a single parent reported significantly higher levels of deviant behavior (F(1, 2327) = 18.19, *p* < 0.001, partial ηp^2^ = 0.01), physical aggression (F(1, 2327) = 9.37, *p* < 0.001, partial ηp^2^ = 0.00), anger (F(1, 2327) = 22.94, *p* < 0.001, partial ηp^2^ = 0.01), hostility (F(1, 2327) = 20.27, *p* < 0.001, partial ηp^2^ = 0.00), and verbal aggression F(1, 2327) = 3.06, *p* < 0.001, partial ηp^2^ = 0.00). Adolescents living in intact families reported higher levels of communication with mother (F(1, 2327) = 4.82, *p* < 0.001, partial ηp^2^ = 0.02), communication with father (F(1, 2327) = 62.56, *p* < 0.001, partial ηp^2^ = 0.03) and perceived family support (F(1, 2327) = 62.22, *p* < 0.001, partial ηp^2^ = 0.03) compared to the ones living in a non-intact family or single-parent or extended family.

### 3.2. Single Group Analyses

As shown in Figure 2, the following variables were used in the model: family structure (intact vs. single parent), deviance propensity (defined by deviant behavior, moral disengagement, and physical aggression), family climate (defined by perceived family support, communication with mother, and communication with father), and anger dysregulation (defined by anger, hostility and verbal aggression). The model in which family structure had both direct and indirect relationships with deviance propensity through family climate and anger dysregulation was an overall good fit to the data (χ^2^ = 327.60, df = 30, *p* = 0.000). Comparative fit indices were above or approached the good fit (CFI = 0.950, TLI = 0.925), while absolute indices were good (RMSEA = 0.065; SRMR = 0.056). All factor loadings were statistically significant and never smaller than 0.53. The average extracted variance (AVE), a measure of the amount of variance that was accounted for by a construct relative to the total amount of variance in its indicators, was 0.50, 0.57, and 0.49 for deviance propensity, anger dysregulation, and family climate, respectively. Because an AVE ≥ 0.50 is recommended to support the convergent validity of the indicator variables, we concluded that deviance propensity, anger dysregulation, were measured with a sufficient degree of accuracy, with family climate closely approaching the threshold.

**Figure 2 ijerph-17-09257-f002:**
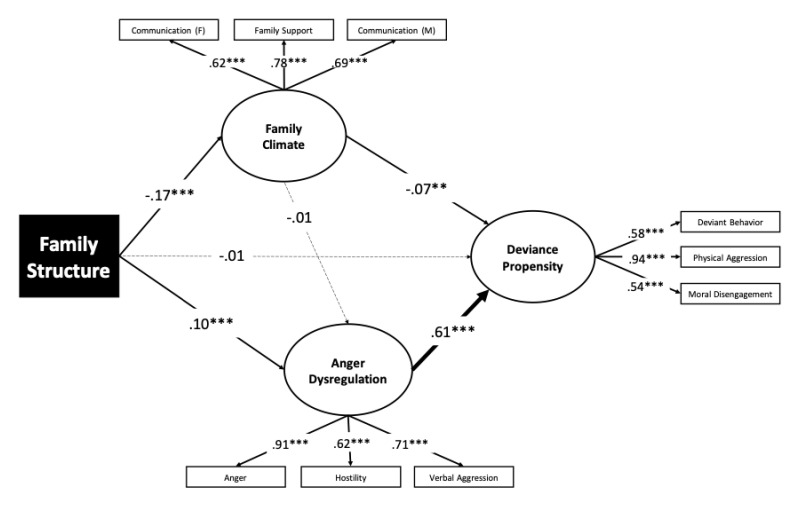
Mediation Analysis. The solid lines represent statistically significant paths (*p* < 0.05). The dashed lines represent not significant paths. The thickness of the lines representing structural relationships is proportional to the effect size of the path coefficient. *** *p* < 0.001, ** *p* < 0.01.

As shown in Figure 2, family structure was associated with family climate negatively (i.e., adolescents raised in single-parent families had worse communication with their parents and perceived to be less supported by their family than adolescents raised in intact families). Family structure was also associated with anger dysregulation, while the path from family structure to deviance propensity was null. Likewise, the family climate was unrelated to anger dysregulation. By contrast, the family climate was significantly linked with deviance propensity (i.e., adolescents who reported better communication with their parents and perceived to be more supported by their family also reported lower deviant behaviors, fewer episodes of physical aggression, and less moral disengagement). The strongest path in the model was for the relationship between anger dysregulation and deviance propensity. To test which of the hypothetical indirect relationships of family structure with deviance propensity was more empirically-supported, we examined the statistical significance of the product of coefficients reported in Figure 2. The indirect associations through family climate (*a* × *b* = 0.03; 95% CI (0.06, 0.01); *p* = 0.009) and through anger dysregulation (*a* × *b* = 0.18; 95% CI (0.24, 0.06); *p* = 0.001) were significant. Because the partially standardized products of coefficients represent an effect size, we appraised the indirect relationship of family structure with deviance propensity as small and small-medium through family climate and anger dysregulation, respectively (z = 0.01 and z = 0.06).

### 3.3. Multigroup Analyses

Next, we examined whether the model parameters could be different for girls and boys using a multigroup analysis. The baseline model was a good fit (χ^2^ = 322.75, df = 60, *p* = 0.000; CFI = 0.957, TLI = 0.936; RMSEA = 0.063, SRMR = 0.057) and served for testing the invariance of the measurement model in subsequent analysis. The model in which we constrained the factor loadings to be equal between girls and boys (χ^2^ = 355.44, df = 66, *p* = 0.000) yielded good comparative fit indices (CFI = 0.953, TLI = 0.935) and absolute indices (RMSEA = 0.062; SRMR = 0.060). The chi-square difference test between the two models was statistically significant (χ^2^ = 32.69, df = 6, *p* = 0.000); however, both the difference in CFI (ΔCFI = 0.001) and NCI (ΔNCI = 0.000) were negligible. These findings were consistent with the standards for establishing measurement invariance of psychometric scales [103,104], supporting the view that the latent variables were measured with the same unit in the two groups. This is an assumption needed to compare the path coefficients between girls and boys in the subsequent analysis.

As one can see from Figure 3, the significant paths in the model for boys linked family structure with family climate and this latter with deviance propensity. While the family structure was not associated with anger dysregulation, there was a marginally significant relationship between family structure and deviance propensity (*p* = 0.055) (i.e., a direct effect). The family climate was unrelated to anger dysregulation. As in single group analyses, anger dysregulation was strongly associated with deviance propensity. Different from boys, for girls family structure, was significantly associated with anger dysregulation (Figure 3), while the direct relation of family structure with deviance propensity was not statistically significant, as also described in single group analysis (Figure 1). In addition, family structure was significantly related to anger dysregulation. Regarding indirect effects, family structure was related to deviance propensity through family climate in boys (*a* × *b* = 0.04; 95% CI (0.08, 0.00); *p* = 0.045) and girls (*a* × *b* = 0.06; 95% CI (0.09, 0.02); *p* = 0.000). The two indirect effects did not differ statistically (*p* = 0.542); however, the partially standardized effect size was larger for girls than for boys (z = 0.02 and z = 0.01, respectively). The most notable difference between girls and boys was for the indirect relationship of family structure to deviance propensity through anger dysregulation, which was statistically significant only for girls (*a* × *b* = 0.18; 95% CI (0.28, 0.08); *p* = 0.000).

**Figure 3 ijerph-17-09257-f003:**
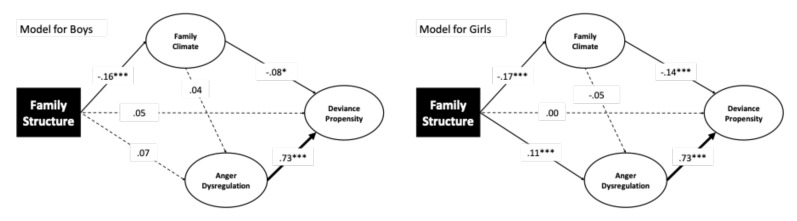
Multigroup analysis of the model tested with gender as a grouping variable. Notes: *** *p* < 0.001; * *p* < 0.05.

### 3.4. Controlling for Socioeconomic Status

As the socioeconomic status of the family can be associated with the constructs included in the present study, we examined whether the SES index was correlated with the family structure and the empirical indicators used to define the latent variables in the model. SES was positively associated with perceived family support (*r* = 0.07; *p* = 0.003) and communication with the father (*r* = 0.05; *p* = 0.014), and negatively with anger (*r* = −0.05; *p* = 0.025) and hostility (*r* = −0.07; *p* = 0.001). Notwithstanding small effect sizes for these correlations, we found it useful to control the effect of socioeconomic status in the model. In the single-group analysis, the model in which we added direct paths from SES to the latent variables depicted in Figure 2, and the covariance of SES with family structure was an overall good fit (χ^2^ = 340.51, df = 36, *p* = 0.000; CFI = 0.949, TLI = 0.923; RMSEA = 0.060; SRMR = 0.052). Consistent with the bivariate correlation analysis, higher SES was associated with better family climate (B = 0.003; SE = 0.001; β = 0.07; *p* = 0.006) and lesser anger dysregulation (B = −0.005; SE = 0.001; β = −0.07; *p* = 0.006). The model’s other parameters did not differ ostensibly from those reported in Figure 2. The indirect relationships were also not affected. In the multigroup-group analysis, the model’s fit was also good (χ^2^ = 376.00, df = 78, *p* = 0.000; CFI = 0.952, TLI = 0.933; RMSEA = 0.057; SRMR = 0.056). As shown in Figure 4, SES was marginally significantly associated with deviance propensity in the female group. Conversely, in the male group, higher SES was significantly associated with better family climate and lesser anger dysregulation. The model’s other parameters did not differ ostensibly from those reported in Figure 3. Likewise, the indirect relationships resembled those previously estimated. Family structure was related to deviance propensity through family climate in boys (*a* × *b* = 0.04; 95% CI (0.09, 0.00); *p* = 0.013) and girls (*a* × *b* = 0.06; 95% CI (0.09, 0.03); *p* = 0.001); z = 0.01 and z = 0.03, respectively. Family structure was associated with deviance propensity through anger dysregulation for girls (*a* × *b* = 0.19; 95% CI (0.28, 0.08); *p* = 0.001) with a larger effect size than for boys (*a* × *b* = 0.17; 95% CI (0.28, 0.08); *p* = 0.039); z = 0.05 and z = 0.08, respectively.

Taken together, socioeconomic status did not confound the relationship between family structure and the other latent variables.

**Figure 4 ijerph-17-09257-f004:**
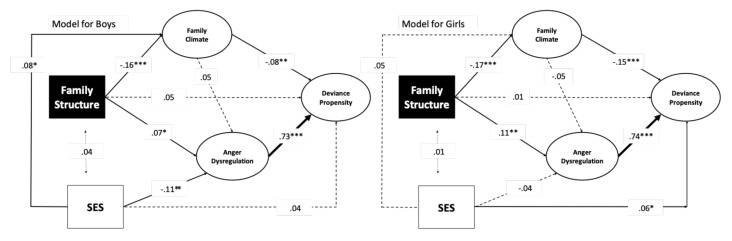
Multigroup analysis of the model tested with gender as a grouping variable, controlling for SES. Notes: *** *p* < 0.001; ** *p* < 0.01; * *p* < 0.05

## 4. Discussion

Our study focused on deviance propensity in adolescence, considering the mediating role of family climate and anger dysregulation in the relationship between family structure and deviance propensity. This hypothesis was supported by some criminological and developmental studies [29,30,31]. The factors evaluated were family structure—divided into intact and single-parent families—(especially living with single mothers); family climate—concerning mother–child, father–child and perceived family support [90,93]; anger dysregulation—composed by three scales of the aggression questionnaire [87,88] anger, hostility, and verbal aggression; deviance propensity, composed by deviant behavior questionnaire [86], moral disengagement scale [94] and the scale of physical aggression of the aggression questionnaire [87,88], and socioeconomic status [98]. According to the construct of deviance’s developmental perspective [18], we conceptualized it as a tendency to commit physically aggressive and illegal acts and to use cognitive mechanisms of moral disengagement to reduce the sense of guilt and the shame produced by the confrontation with society [25]. Anger dysregulation was identified as an inability to manage emotional (anger), cognitive (hostility), and behavioral (verbal aggression) components of anger. The family variables we have considered in the present study were the following: perceived communication in the family, family support, and family structure. One of the strengths of the study is the large sample (*n* = 2328) examined, which is composed of adolescents of both genders aged 13 to 19, mainly of Italian nationality and coming from Southern Italy. The present study tested several hypotheses concerning the relationship between the family system and antisocial tendencies. First of all, concerning the influence of family structure (intact vs. single parent) on positive vs. negative family climate, there was a significant negative relationship between variables: adolescents coming from mono-parental families showed worse communication with their parents and perceived to receive less support from their family compared to adolescents raised in intact families. This pattern is consistent with other research showing that the negative quality of family climate is strongly related to deviance and antisocial conduct [49]. A good family climate facilitates the management of emotions, resilience and is effective in the reduction of deviance, in turn promoting greater well-being [105,106]; on the other side, poor family communication and low support involve the experimentation of an affective and emotional gap, which can be translated into physical violence and aggressive behavior [47,48,49]. Indeed, it could be possible to have one parent and experience a good relationship with them and have both parents, but negative relations with them.

Regarding the relationship between family structure and anger dysregulation, we have detected a significant positive relationship between variables: living with a single parent is a risk factor for anger dysregulation in this particular sample. This was an expected result because we had postulated a protective role of family intactness [40] on anger dysregulation. Family structure was not directly related to deviance propensity. The variable has not a determinant impact like the one, which can be reached by the family climate [107]. Indeed, it could be possible that living with both parents alone is not forcedly a protective factor for the development of deviance propensity. These two latter variables are linked only indirectly through problems with anger regulation and bad quality of communication with parents, and perceived lack of family support [46]. As reported in previous research [49,107], our results showed that the negative quality of family climate is strongly related to deviance and antisocial conduct. Indeed, we have verified that a positive family climate was negatively related to deviance propensity, thus confirming H4a, i.e., the protective role of positive communication with both parents and perceived support of the family on deviance propensity. In fact, poor family communication and low support could lead to emotional issues, physical violence, aggressive behavior [46,47], low self-control and self-efficacy [50,51]. Anger dysregulation was positively associated with Deviance propensity, thus confirming H4b, i.e., anger dysregulation acted as a trigger of deviance propensity and so higher levels of anger, hostility, and verbal aggression were associated with a higher tendency to deviant behaviors, a higher physical aggression attitude, and more moral disengagement. Despite this result being well-known and confirming the main results of international literature, it could be useful to enrich the knowledge in the Italian context and to develop prevention and intervention programs with the aim to improve parent–children relationships. The indirect associations of family climate on deviance propensity through family climate and through anger dysregulation were the only significant association and had a small effect and small-medium effect, respectively.

Regarding the multigroup comparison (H5), aimed to test the invariance of the structural paths and indirect relationships, we demonstrated that the latent variables were measured in the same way in girls and boys. Family structure was associated with deviance propensity through family climate in both girls and boys, but the protective effect was stronger for girls. Unexpectedly, the indirect relationship of family structure with deviance propensity through anger dysregulation was significant only for girls. These results suggest that girls tend to regulate anger by internalizing it, expressing a sense of hostility, a feeling of rage, and showing verbal aggression, instead of acting out or using physical violence like males [108]. Moreover, negative events could impact females differently, especially during childhood and adolescence. For instance, Wallerstein and Blakeslee conducted a study on the long-term outcomes of the divorce in a sample of 60 families (120 parents and 131 children) [109]. The researchers interviewed parents and children at intervals of one year, five years, and ten years to monitor the impact of divorce in their lives. Results showed that girls reacted better emotionally and behaviorally than boys, but in early and late adolescence, they developed what the authors defined as “the sleeper effect”. Indeed, girls showed serious psychological and behavioral issues in their emotional relationships. Instead, boys were more likely to be involved in delinquency and in acting out and aggressive behavior. Similarly, in our research, when comparing the conduct of girls and boys, we found that they have a different relationship with anger, which is expressed through violent physical acts in males, and cognition and negative feelings in girls.

One of the ancillaries aims of this study was to control the potential influence of SES, measured in the present study with the Hollingshead index, on all the variables of the present study. Indeed, a higher SES was positively related to perceived family support and communication with father and negatively related to anger and hostility. These relationships, even if they were characterized by a very small effect size, were in line with the single group analysis that showed that SES was related to positive Family climate and lower levels of anger dysregulation. Regarding gender differences, our multigroup analysis showed that SES was associated with deviance propensity only in girls: higher levels of SES were associated with higher levels of Deviance Propensity. This result is interesting and could be the focus of future studies on the topic. On the other side, in boys, SES was positively associated with family climate: a higher SES was related to better communication and perception of family support. SES was negatively associated with anger dysregulation, showing a protective role of socioeconomic status for males. Summarizing, SES was not a confounder in the relationship between family structure and all the other variables.

One of the main limitations of the study regards its cross-sectional research design, which prevents from formulating any causal inferences, which cannot properly be made without active control over the variables concerned. Longitudinal or experimental studies proving the existence of causal relations are a challenge for future research. A second limitation was that the effect sizes for key correlations, path coefficients (except for the path linking anger dysregulation to deviance propensity), and indirect effects were only small–medium (according to Cohen’s standards). However, the total indirect effects estimated in our model was sufficient to support the claim that family structure, family climate, and anger dysregulation contributed significantly to deviance propensity. A third limitation is that it was not possible to consider other features of family structure, e.g., stepfamilies, cohabiting-couple families or other families such as extended families, rather than intactness and mono-parental families, due to the particular composition of our sample.

The results provided important information on some specific factors, for example, the role of the single-parent family structure in the development of antisocial and aggressive behavior. Another important aspect considered is the division in internalized types of aggression, i.e., anger, hostility, and externalized aggression, such as verbal and physical aggression. Future studies in the Italian context could evaluate family structure’s impact while taking into account different types of structures and the age at which the separation happened, to understand if there are any differences between adolescents who have a single-parent family since they were children and adolescents who lived in a broken family only in adolescence. Furthermore, it could be useful to integrate adolescents’ self-evaluation with parents’ evaluation to investigate parents’ perceptions of children’s well-being and promote interventions in the family context. Finally, a study on teenagers’ disclosure with mother and father could add interesting findings to understand the difference between mother-children and father–children communication.

## 5. Conclusions

In conclusion, this study suggested that family climate and anger dysregulation were both involved in the process that links family structure with deviance propensity. When comparing girls and boys, the indirect effect of family climate was significant for both, mostly for girls. These results are fundamental in planning preventive interventions in family and school settings, to encourage family communication and pay greater attention to families and the quality of the parent–child relationship. Indeed, good communication and support could reduce the risk factors of adolescence and prevent them from conduct issues. Moreover, it could be useful to develop gender-specific interventions for girls to help them in anger regulation. This research is a starting point in understanding the complex phenomenon of adolescent deviance in the Italian context.

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
