# Peer review of "Is Family Structure Associated with Deviance Propensity during Adolescence? The Role of Family Climate and Anger Dysregulation"

_ijerph, 2020, doi:10.3390/ijerph17249257_

Round 1
Reviewer 1 Report
Interesting paper, I suggest using more sub headings. Ie the introductions and lit review run into each other. I think a more detailed lit review around the many constructs included is a must.
The constructs( variables) should be defined again in the results section. As a reader I had a difficult time remembering what all the variables meant.
It was unclear in the study if households with 2 parents meant biological mother and father only? What about same sex households? Adoptive parents? Is the single parent a biological parent? etc. This needs to be clarified.
You often mention many studies etc... you needs to list these studies.
In the ANOVA results section you mention that the results were siginificant, but did not include any p-value.
Finally, the answers to the research questions need to be highlighted at the end of the study.
Line 369 - grammar, ie "who"
Reviewer 2 Report
This study utilized a sample of 2,194 adolescents from Italy to investigate how family climate mediated the relationship between family structure (intact family vs. non-intact family) and adolescents’ deviance propensity. The Structure Equation Modeling (SEM) analysis indicated that family climate significantly mediated the relationship between family structure and deviance propensity for girls but not for boys and the mediating role of anger dysregulation was statistically negligible. Though these findings are not new, studies are limited in the Italian context. As such, the findings are noteworthy. However, there are both theoretical and methodological issues that need to be addressed before this study can be considered for publication in peer-reviewed journals. I include my observations and suggestions below and hope they are helpful for the authors’ revision endeavors.
Theoretical Issues
First, the effects of family structure on adolescents’ wellbeing (including deviance) are well-known but empirical studies conducted in the Italian context are limited. Given this research paucity, it would be interesting for the reader to understand how various family structures (e.g., two biological families, single mother/father families, step-families, cohabiting-couple families or other families such as extended families) might be related to adolescents’ wellbeing, especially deviance, in Italy. In addition, I suggest that the author review the following article for developing more systematical perspectives in their literature review: Lansford, J. E., Ceballo, R., Abbey, A., & Stewart, A. J. (2001). Does family structure matter? A comparison of adoptive, two‐parent biological, single‐mother, stepfather, and stepmother households. Journal of Marriage and family, 63(3), 840-851.
Second, past research suggested that the effects of family structure on adolescents’ wellbeing, including deviance, could be explained away by families’ economic resources. By the same token, parenting alone may not be that important as along as economic resources are taken into consideration. Though the authors correctly hypothesized insignificant and direct relationship between family structure and adolescents’ deviance tendency, they however, didn’t even consider economic resources. How do we know that the mediating role of family climate (this may include parenting in addition to parent-child communication) will not go away when economic resources are controlled for? This can be a potential flaw of the present study. To guard against this potential flaw, please review and consider the following publications: (1) Thomson, E., & McLanahan, S. S. (2012). Reflections on “Family structure and child well-being: Economic resources vs. parental socialization”. Social Forces, 91(1), 45-53; (2) McLanahan, S. (1985). Family structure and the reproduction of poverty. American journal of Sociology, 90(4), 873-901; and (3) McLanahan, S., & Percheski, C. (2008). Family structure and the reproduction of inequalities. Annu. Rev. Sociol, 34, 257-276. So, please include and analyze economic resources.
Third, the authors didn’t provide theoretical reasons for why there would be gender differences in their conceptual models. In fact, after their multigroup analysis, the author didn’t even speculate why the mediating models only worked for girls but not for boys. Are the observed gender differences due to (1) the nature of deviance, (2) differences in parent-child communication and thus family socialization processes, or (3) gendered effects of family structure on family climate as well as differential patterns in deviance? Pleas find possible and useful arguments made by Wallerstein (see Wallerstein, J. S., & Blakeslee, S. (1989). Second chances: Men, women and children a decade after divorce. New York: Ticknor & Fields). More work is need to explain possible gender differences.
Methodological Issues
One major concern is that the authors didn’t provide enough information about their study design. For example, I still didn’t know if this was a random sample. If not, how the high-school students were selected? How representative was this sample with reference to various regions in Italy (e.g., .25.6% from the Center of Italy, 10% from the North and 64.4% from the South)?
The dummy-coded exogenous family structure variable could be limited. Did the author consider single parent, especially single-mom, families as compared with intact families? The current dummy variable may mask the effects of single parent families that have been frequently studied and emphasized in the literature.
A minor terminology issue needs attention. The authors occasionally used the term “youth” interchangeably with adolescents. This is not appropriate. According to WHO, adolescents’ age range is from 10 to 19, whereas youth’s age range is from 15 to 24. Consistent use of adolescents is suggested.
Round 2
Reviewer 2 Report
Overall, the authors made some good-faith efforts to revise the manuscript and they were responsive to my comments. However, in the revised texts (marked yellow), they got very sloppy. There were so many errors, both grammatical and substantive (please see a few examples included below). In addition, I strongly urge the authors to use editing service to revised and line-edit the revised text. It was very difficult for me to read through.
Why reliability coefficient for Anger is so low? .54. Please explain.
For the Hollingshead SES Index, please say “occupational prestige” rather than “the professional status”.
The following sentence is wrong (lines 415-417).
“Adolescents living in intact families reported higher levels of communication with mother (F(1, 327) = 3.66, p < 0.001, partial ηp2= .02), communication with father (F(1, 2327) = 61.94, p < 0.001, partial ηp2 = .03) and family support (F(1, 2193) = 56.64, p < 0.001, partial ηp2 = .00) compared to the ones living in an intact family.”
Please change “compared to the ones living in an intact family” to “compared to those who reside in a non-intact family or single-parent or extend family”.
Why in Figure 2 was the Multi-Group analysis referred to as “CFA”? the authors didn’t conduct a Confirmative Factor Analysis (CFA), instead, the y conducted a structural equation modeling (SEM) with latent variables.
Line 26, omit “indirectly”; mediating implies indirect association.
Lines 153-154, change “marriages last less” to the length of marriage is shorter”.
Lines 158-59, the phrase “but in some cases (if parental health is monitored) even higher emotional well-being” is confusing. What does it mean by higher well-being? Better off or worse off?
Line 184, “The SES is simultaneously a cultural and an economic index”? Really? The SES is often regarded as structural rather than cultural.
